# Emergence of curved light-cones in a class
# of inhomogeneous Luttinger liquids

**Jérôme Dubail[1][*], Jean-Marie Stéphan[2] and Pasquale Calabrese[3]**

**1** CNRS & IJL-UMR 7198, Université de Lorraine, F-54506 Vandoeuvre-lès-Nancy, France
**2** Univ Lyon, Université Claude Bernard Lyon 1, CNRS UMR 5208, Institut Camille Jordan,
43 blvd. du 11 novembre 1918, F-69622 Villeurbanne cedex, France
**3** SISSA and INFN, Via Bonomea 265, 34136 Trieste, Italy

[*] jerome.dubail@univ-lorraine.fr

## Abstract

The light-cone spreading of entanglement and correlation is a fundamental and ubiquitous feature of homogeneous extended quantum systems. Here we point out that a class of inhomogenous Luttinger liquids (those with a uniform Luttinger parameter $K$) at low energy display the universal phenomenon of curved light cones: gapless excitations propagate along the null geodesics of the metric $ds^2 = dx^2 - v(x)^2 dt^2$, with $v(x)$ being the calculable spatial dependent velocity induced by the inhomogeneity. We confirm our findings with explicit analytic and numerical calculations both in- and out-of-equilibrium for a Tonks-Girardeau gas in a harmonic potential and in lattice systems with artificially tuned hamiltonian density.


## Contents

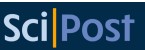

# 1   Introduction

Light-cone propagation of signals in condensed matter and cold atom physics has attracted a lot of attention in the past decade. Contrarily to what happens in relativistic high-energy physics, where no signal can propagate faster than the speed of light, in non-relativistic quantum systems a maximal speed of propagation does not necessarily exist. A very profound result of mathematical physics is that excitations in discrete models with a finite-dimensional local Hilbert space and with local interactions have indeed a bounded velocity $v_{max}$ [1]. This classic result by Lieb and Robinson has had many ramifications over the years [2]. Roughly, it implies that when these systems are perturbed at a given position $x_0$, after a time $t$ the signal generated by the perturbation cannot propagate farther than $x_0 \pm v_{max}t$. The existence of a maximum speed of propagation has deep consequences in the non-equilibrium dynamics of these systems, resulting in the so called light-cone spreading of correlations [3–7] and entanglement [8–23, 28] e.g. following a global or a local quantum quench, as reviewed e.g. in [34, 35]. Such light-cone spreading has been also experimentally observed [36–38]. Furthermore, the presence of a velocity bound also affects the transport properties of these systems [39–41].

Apart from the Lieb-Robinson rigorous mathematical bound, condensed matter systems sometimes display the same behavior as relativistic models of high-energy physics: this happens at quantum critical points with dynamic exponent $z = 1$, where emergent massless quasi-particles propagate at a critical (sound) velocity $v$ which plays the same role as the speed of light in high-energy physics. This relativistic behavior *emerges* only when probing the low-energy (large distance) universal features of these systems. In particular in one spatial dimension, these systems mainly fall in the Luttinger liquid paradigm, as pointed out long ago by Haldane [42]. A wide class of one-dimensional systems such as spin chains, one-dimensional superfluids, electrons in one-dimensional wires, or clouds of ultra-cold atoms in tightly elongated traps are indeed well described by Luttinger liquid theory.

In the limit of large distances and time separations, all Luttinger liquids in equilibrium display power law singularities in time-dependent correlators (such as the Green function) on the light cone. This is a straightforward manifestation of the light-cone propagation of signal at low energy. However, this is more cleanly devised in non-equilibrium situations such as a local quench following a cut-and-glue protocol [21–23, 28]. Quite remarkably, since a local quench only probes the low-energy part of the many-body spectrum, the light-cone spreading of entanglement and correlations can be observed in general Luttinger liquids, even in those systems that have unbounded velocities and do not satisfy Lieb-Robinson [28].

The purpose of this paper is to investigate light-cone propagation in certain inhomogeneous quantum critical systems in 1d. We focus on two particular microscopic realizations of an inhomogeneous Luttinger liquid: impenetrable bosons in a non-uniform trap potential, and spin chains with position-dependent couplings between neighboring spins. These two systems share a crucial property, on which our entire analysis is relying: the Luttinger parameter $K$ is constant in those systems. This will be explained in more details shortly; it is important,

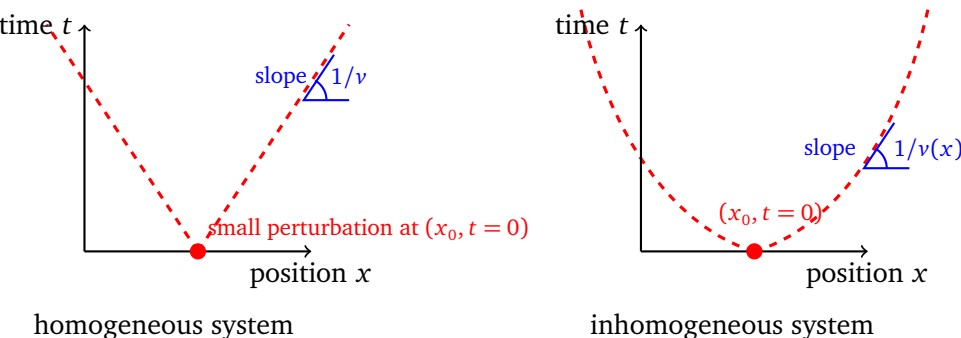

Figure 1: Cartoon of the phenomenon investigated in this paper. (Left) When one applies a small perturbation to a Luttinger liquid at position $x_0$ and at time $t = 0$, the signal generated by the perturbation propagates along the left- and right-arms of the light-cone, $x = x_0 \pm vt$. (Right) in an inhomogeneous system, the velocity $v$ of gapless excitations becomes a function of position, and, as a result, the propagation of gapless excitation is along curved light-cones. The goal of this paper is to confirm the emergence of such curved light cones in some concrete microscopic models.

however, to emphasize that there exist other systems, also in the Luttinger liquid universality class, which would not have a constant Luttinger parameter $K$ in inhomogeneous situations, to which our analysis does not apply (the most remarkable example being the 1d delta Bose gas in a non-uniform trap potential, away from the Tonks-Girardeau limit, see e.g. Ref. [29]).

Our calculations are based on the recently introduced framework to describe gapless inhomogeneous systems by means of conformal field theories (CFT) in a curved background [43], see also [44–46]. Our main conclusion will be that, in those systems that are inhomogeneous, but have a constant Luttinger parameter $K$, the velocity of gapless excitations $v$, analogous to the speed of light, is position-dependent, $v \to v(x)$. This has a dramatic consequence on light-cone propagation: the light-cones become curved, as illustrated in Fig. 1. Our approach allows to quantitatively characterize the curvature of the light-cone, i.e. to determine analytically the explicit space dependence of the sound velocity. Similar curved light-cones have also been observed in time-dependent quenches [30–33], where the velocity depends on time but not position. In the present work we focus on space-dependent velocities instead.

The paper is organized as follows. In Section 2 we show how the inhomogeneous Luttinger liquid naturally appears when describing a gapless 1d quantum system and we introduce a few models with constant Luttinger parameter $K$, as well as all other assumptions exploited in our calculation. In Section 3 we apply the framework of CFT in curved background to the Tonks-Girardeau gas in a harmonic trap both in equilibrium and for a local quench. In Section 4 we present our results for lattice models with inhomogeneous hamiltonian density. Finally in Section 5 we draw our conclusions and discuss some open problems.

## 2 Inhomogeneous Luttinger liquid with constant Luttinger parameter $K$

The Luttinger liquid is an effective description of systems of particle or spins that is valid on distance- and time-scales that are large compared to microscopic ones (for instance, the lattice spacing, or the interparticle distance). Here we quickly introduce the formalism that we will apply later to some specific microscopic models. Only the aspects of Luttinger liquid physics

that are strictly necessary to this paper will be discussed; moreover, our treatment is slightly non-standard, because our goal is to emphasize the role of the inhomogeneity of the model. For a more standard introduction and thorough discussion of Luttinger liquid physics, we refer the reader to the textbooks [47] and [48].

## 2.1 The height field

Imagine that we have a microscopic model of mobile particles on the real line, at positions $x_1, x_2, \ldots, x_N$, and that we are interested in describing correlations of the density operator $\rho(x) \equiv \sum_j \delta(x - x_j)$. A configuration of particles can be represented by the height function $h(x)$, defined such that

$$\rho(x) = \frac{1}{2\pi} \partial_x h(x). \tag{1}$$

The factor $2\pi$ is introduced for notational convenience. Notice that this relation defines $h(x)$ only up to a constant shift $h(x) \to h(x) + \text{const}$. If we know how to calculate correlation functions of the height field, then we know how to calculate correlation functions asymptotically in the microscopic model. This is the basic idea of both Luttinger liquid theory and the Coulomb gas approach to 2d statistical mechanics [49]. The functions $\rho(x)$ and $h(x)$ represent the configuration of the system at a given instant, say $t = 0$. The time dependence is easily written in terms of the Hamiltonian $H$ of the microscopic model

$$\rho(x, t) = e^{iHt} \rho(x) e^{-iHt}, \qquad h(x, t) = e^{iHt} h(x) e^{-iHt}, \tag{2}$$

or in imaginary time

$$\rho(x, \tau) = e^{\tau H} \rho(x) e^{-\tau H}, \qquad h(x, \tau) = e^{\tau H} h(x) e^{-\tau H}. \tag{3}$$

For instance, the two-point function $\langle \psi_0 | h(x, \tau) h(x', \tau') | \psi_0 \rangle$, in the ground state $|\psi_0\rangle$ of $H$, is defined as the (imaginary-) time-ordered expression

$$\langle \psi_0 | h(x, \tau) h(x', \tau') | \psi_0 \rangle = \lim_{T \to \infty} \begin{cases} \frac{\text{tr}[e^{-(T-\tau)H} h(x) e^{-(\tau-\tau')H} h(x') e^{-\tau'H}]}{\text{tr}[e^{-TH}]} & \text{if} \quad \tau > \tau' \\ \frac{\text{tr}[e^{-(T-\tau')H} h(x') e^{-(\tau'-\tau)H} h(x) e^{-\tau H}]}{\text{tr}[e^{-TH}]} & \text{if} \quad \tau' > \tau. \end{cases} \tag{4}$$

We have a height field $h(x, \tau)$ that lives in 2d spacetime; the goal is now to write an action $\mathscr{S}[h]$ for this field, that can be used to calculate correlation functions.

## 2.2 Coarse-grained action for the height field, and emergence of the gaussian free field

We now assume that the Hamiltonian $H$ of the microscopic model is local, and we adopt a coarse-grained perspective. Because the model is local and invariant under constant shifts $h(x, \tau) \to h(x, \tau) + \text{const}$, the fluctuations of $h$ must be captured by a local action of the form

$$\mathscr{S}[h] = \int \mathscr{L}(\partial_x h, \partial_\tau h, \partial_x^2 h, \partial_x \partial_\tau h, \partial_\tau^2 h \ldots) \, dx \, d\tau. \tag{5}$$

One assumes for simplicity that there is a unique classical configuration $h_{\text{cl.}}$ that minimizes the action (5), up to constant shifts of $h$. Then, substituting $h \to h + h_{\text{cl.}}$, one can expand around $h_{\text{cl.}}$, which gives

$$\mathscr{S}[h + h_{\text{cl.}}] - \mathscr{S}[h_{\text{cl.}}] =$$
$$\int \frac{1}{2} \left( \frac{\partial^2 \mathscr{L}}{\partial (\partial_x h)^2} (\partial_x h)^2 + 2 \frac{\partial^2 \mathscr{L}}{\partial (\partial_x h) \partial (\partial_\tau h)} (\partial_x h)(\partial_\tau h) + \frac{\partial^2 \mathscr{L}}{\partial (\partial_\tau h)^2} (\partial_\tau h)^2 \right) dx \, d\tau$$
$$+ \text{ higher order terms.} \tag{6}$$

All higher order terms have scaling dimension larger than 2, and are irrelevant in two dimensions. This is the fundamental reason for the universality of the Luttinger liquid model: under very general assumptions, the underlying field theory turns out to be a free, gaussian, field theory, with the quadratic action (6). There are only three adjustable parameters in this action, namely the three independent entries of the hessian $\nabla^2 \mathscr{L}$ at the point $h = h_{\text{cl.}}$. It is convenient to interpret the inverse of the hessian as a metric tensor

$$
g \equiv \begin{pmatrix} \frac{\partial^2 \mathscr{L}}{\partial(\partial_x h)^2} & \frac{\partial^2 \mathscr{L}}{\partial(\partial_x h)\partial(\partial_\tau h)} \\ \frac{\partial^2 \mathscr{L}}{\partial(\partial_x h)\partial(\partial_\tau h)} & \frac{\partial^2 \mathscr{L}}{\partial(\partial_\tau h)^2} \end{pmatrix}^{-1}.
$$

Then the action (6) becomes, with $(\mathrm{x}^1, \mathrm{x}^2) = (x, \tau)$,

$$
\Delta \mathscr{S}[h] = \mathscr{S}[h + h_{\text{cl.}}] - \mathscr{S}[h_{\text{cl.}}] = \frac{1}{8\pi K} \int d^2 \mathrm{x}\, g^{ij} (\partial_i h)(\partial_j h) \sqrt{g}, \tag{7}
$$

where $\frac{1}{K} \equiv 4\pi \sqrt{\det(\nabla^2 \mathscr{L})}$. In principle, the value of $K$ may depend on position x, however in this paper, we will assume that it does not. Below, we will discuss in more details in which type of microscopic models this assumption is valid. The case of non-constant $K$ is also very interesting (and it has appeared, in Hamiltonian form, for some particular choices of the function $K(\mathrm{x})$ in Refs. [29, 50–52], but in general it is more complicated, and it will be studied elsewhere).

The key point, which allows to solve the Luttinger liquid model exactly, is that the action (7) is nothing but a gaussian free field in the conformal class of the metric $[g]$; indeed the action (7) is invariant under Weyl transformations $g \to e^{2\Phi} g$, and thus depends on the conformal class of $g$, noted $[g]$, rather than $g$ itself. Solving the model then boils down to finding its propagator $\langle h(\mathrm{x}) h(\mathrm{x}') \rangle$, which is an easy exercise in conformal mapping and potential theory. We will come back to this below, but first, let us introduce and discuss the two additional assumptions that we make throughout this paper.

## 2.3 Our assumptions

Throughout this paper, we will be dealing with microscopic systems in the Luttinger liquid universality class, that satisfy the following three assumptions.

**Assumptions in this paper:**

1. the parameter $K$ is uniform, i.e. it does not depend on position $(x, \tau)$ in spacetime; consequently, large-scale correlations can be obtained from the action (7);

2. we consider small perturbations around the ground state of $H$; consequently, the entries of the metric tensor $g$ do not depend on time $\tau$ (but they do depend on position $x$);

3. the system is time-reversal invariant; consequently the component $g_{x\tau}$ of the metric tensor (which transforms as $g_{x\tau} \to -g_{x\tau}$ under time-reversal symmetry) is zero.

The first one, namely the fact that $K$ is uniform, has already been mentioned, and is further discussed in the next section. For now, let us focus on the consequences of assumptions 2 and 3. Because of Weyl invariance of (7) only the two ratios $g_{x\tau}/g_{xx}$ and $g_{\tau\tau}/g_{xx}$ play a role in the model,

$$
ds^2 = g_{ij} dx^i dx^j \propto dx^2 + 2 \frac{g_{x\tau}}{g_{xx}} dx\, d\tau + \frac{g_{\tau\tau}}{g_{xx}} d\tau^2. \tag{8}
$$

Then, as a consequence of assumptions 2 and 3, we have

$$
ds^2 = dx^2 + v^2(x) d\tau^2, \tag{9}
$$

where we defined $v^2(x) \equiv g_{\tau\tau}(x)/g_{xx}(x)$. Eq. (9) is very simple, yet it is the most profound equation of the paper. We will see that gapless excitations automatically propagate along the null geodesics of this metric (after Wick rotation), which lead to the appearance of the curved light-cones illustrated in Fig. 1. It is worth emphasizing that our assumptions 2 and 3 are merely technical ones; while they simplify our task considerably in the following, they are in principle not crucial. One could easily relax them without affecting our main conclusion, which is the emergence of curved light-cones. One particular example where the entries of the metric $g_{ij}$ are all non-zero and depend both on position and time is the one of the quench from a domain-wall initial state, which was treated in Refs. [43,44].

Throughout the paper, it will be useful to use the coordinate

$$\tilde{x} \equiv \int_{x_L}^{x} \frac{du}{v(u)}, \tag{10}$$

where $x_L$ is some reference point, that we will take to be the position of the left boundary of the system. Physically, $\tilde{x}$ is the time that is needed by a signal that propagates at the local velocity $v(x)$ to travel from the boundary at position $x_L$ to position $x$. The metric (9) then takes the form

$$ds^2 = v^2(x)\left[d\tilde{x}^2 + d\tau^2\right] = e^{2\sigma}dzd\bar{z}, \tag{11}$$

where $e^{\sigma} = v(x)$ and $(z,\bar{z}) = (\tilde{x} + i\tau, \tilde{x} - i\tau)$. In these coordinates, the action (7) takes the standard form

$$\Delta\mathscr{S}[h] = \frac{1}{2\pi K}\int (\partial_z h)(\partial_{\bar{z}} h)dzd\bar{z}. \tag{12}$$

## 2.4 Discussion: why $K$ is uniform for a certain subset of Luttinger liquids

We have explained that we restrict our analysis to the case of systems with constant Luttinger parameter $K$; let us now emphasize that, even though this is a strong restriction, there are still various interesting microscopic models that satisfy this assumption.

**Models that map to free fermions: $K = 1$.** The first important class of microscopic models with constant $K$ is the ones that map to free fermions. Relevant examples include models of hard-core boson [53] or hard-core anyons [54], either in the continuum or on the lattice and spin-chains such as the XX model. Let us briefly recall why, in these models, one necessarily has $K = 1$, ruling out the possibility of a varying Luttinger parameter $K$.

Since the model has underlying non-interacting fermionic degrees of freedom, the theory described by the action (12) must contain an operator that creates a fermionic excitation, that evolves freely. Since we are in an effective theory with no mass, the free fermion excitation must be right- or left-propagating, with velocity $\pm v$ (or $\pm v(x)$ in the inhomogeneous case with position-dependent velocity). To construct an operator that creates left- and right-moving excitations, one proceeds as follows. The propagator derived from the action (12) satisfies the equation $\Delta \langle h(z_1,\bar{z}_1)h(z_2,\bar{z}_2)\rangle = -4\partial_{z_1}\partial_{\bar{z}_1}\langle h(z_1,\bar{z}_1)h(z_2,\bar{z}_2)\rangle = 4\pi K \delta^{(2)}(z_1 - z_2)$, so it must behave locally as

$$\langle h(z_1,\bar{z}_1)h(z_2,\bar{z}_2)\rangle \underset{z_1 \to z_2}{\sim} -K\log|z_1 - z_2|^2. \tag{13}$$

Equivalently, one can think of $h(z,\bar{z})$ as being the sum of chiral and anti-chiral components

$$h(z,\bar{z}) = \sqrt{K}\left[\varphi(z) + \bar{\varphi}(\bar{z})\right], \tag{14}$$

which have the following propagators

$$\langle \varphi(z)\varphi(z')\rangle \sim -\log(z - z'), \qquad \langle \overline{\varphi}(\bar{z})\overline{\varphi}(\bar{z}')\rangle \sim -\log(\bar{z} - \bar{z}'), \qquad \langle \varphi(z)\overline{\varphi}(\bar{z}')\rangle \sim 0, \tag{15}$$

for consistency of (14) with (13). The operators that create chiral excitations are then of the form

$$e^{i\alpha\varphi(z)} \qquad \text{and} \qquad e^{-i\beta\overline{\varphi}(\bar{z})}, \tag{16}$$

where the exponential is normal ordered. These operators create one particle, so they must have charge one, by definition. This means that the coefficient of their operator product expansion with the density operator $\rho(z,\bar{z}) = \frac{\sqrt{K}}{2\pi}[\partial\varphi + \partial_{\bar{z}}\overline{\varphi}]$, i.e.

$$\rho(z_1,\bar{z}_1) \cdot e^{i\alpha\varphi(z_2)} \underset{z_1 \to z_2}{\sim} i\frac{\sqrt{K}\alpha}{2\pi}e^{i\alpha\varphi(z_2)}, \qquad \rho(z_1,\bar{z}_1) \cdot e^{-i\beta\overline{\varphi}(\bar{z}_2)} \underset{\bar{z}_1 \to \bar{z}_2}{\sim} -i\frac{\sqrt{K}\beta}{2\pi}e^{-i\beta\overline{\varphi}(\bar{z}_2)}, \tag{17}$$

must be $\frac{i}{2\pi}$ and $\frac{-i}{2\pi}$ respectively. This implies

$$\alpha = \beta = \frac{1}{\sqrt{K}}. \tag{18}$$

Similarly, by looking at the operator product expansion of $e^{i\alpha\varphi(z)}$ and $e^{-i\alpha\varphi(z)}$ with the stress-tensor, one gets their spin, which is equal to

$$e^{i\frac{\varphi(z_2)}{\sqrt{K}}}: \quad \text{spin} \quad \frac{1}{2K}, \qquad\qquad e^{-i\frac{\overline{\varphi}(\bar{z}_2)}{\sqrt{K}}}: \quad \text{spin} \quad -\frac{1}{2K}. \tag{19}$$

But these operators are fermions, so they must have spin $\pm\frac{1}{2}$. Thus we arrive at

$$K = 1 \qquad\qquad \text{(for free fermions)} \tag{20}$$

as claimed.

**Models with artificially varying hamiltonian density.** Models with underlying free fermionic degrees of freedom are not the only ones that possess a fixed parameter $K$. If a Luttinger liquid is SU(2) symmetric, then it necessarily has $K = 1/2$, for reasons that are similar to the ones we just explained for the free fermion case. Namely, SU(2) symmetry implies that the massless left- and right-excitations can be separately organized in SU(2) multiplets. For this to be possible, one must find chiral operators that act as the generators $S^{\pm}$, $S^z$ of the SU(2) algebra, up to central extension. These three current operators are of the form $e^{\pm i\frac{\varphi}{\sqrt{K}}}$, and they must have scaling dimension 1 (or, equivalently, spin 1), and this fixes $K = 1/2$. Thus, we could think of investigating inhomogeneous microscopic models with SU(2) symmetry and emergent Luttinger liquid physics. It turns out that such microscopic models are very constrained. To see this, let us start from the *homogeneous* antiferromagnetic Heisenberg spin-1/2 chain

$$H = J\sum_x \mathbf{S}_x \cdot \mathbf{S}_{x+1}, \qquad J > 0, \tag{21}$$

with $\mathbf{S} = (S^x, S^y, S^z)$. It is well-known that the large-distance, low-energy, behavior of this model is captured by the Luttinger liquid model with $K = 1/2$.

Now we ask: how can we make this model *inhomogeneous*, while preserving SU(2) symmetry, such that the large-scale behavior will be captured by an inhomogeneous Luttinger liquid with constant $K = 1/2$?

The simplest answer is obviously: vary the coupling $J$ with position, $J \to J(x)$. If $J_j$ varies slowly enough with position $x$, then we will indeed have a model that is locally like a homogeneous Luttinger liquid, but since the velocity $v$ is proportional to the coupling $J$, it will naturally vary with position. Another possibility would be to include position-dependent next-nearest neighbor terms $\mathbf{S}_x \cdot \mathbf{S}_{x+2}$. Such terms would also affect the local velocity; in addition, they could also introduce perturbations by operators that are marginally relevant or

irrelevant, depending on the sign of the coupling (in the marginally relevant case, this would lead to dimerization, so it would destroy the long-range Luttinger liquid physics in which we are interested).

Thus, we arrive at the conclusion that, imposing SU(2) symmetry in order to protect the value of the Luttinger parameter $K$, the simplest possible inhomogeneous deformation of the Heisenberg Hamiltonian is to vary the coupling $J$ with position.

A similar conclusion holds for supersymmetric models.[1] The superysmmetric spin chain of Refs. [55–57] falls in the Luttinger liquid class with $K = 1/3$. Lattice supersymmetry is reflected in the continuous field theory in the fact that, at this particular value of $K$, there are two chiral operators $e^{\pm i\sqrt{3}\varphi}$ (and anti-chiral ones $e^{\pm i\sqrt{3}\overline{\varphi}}$) with conformal dimension 3/2, that generate an $\mathcal{N} = (2, 2)$ supersymmetry algebra; see Ref. [55] for a detailed study of this model. If one tries to find an inhomogeneous deformation of this model while keeping supersymmetry, such that the value of $K$ remains protected, then one finds that the simplest deformation allowed is also just to vary the amplitude of the Hamiltonian density with position.

Hence, we are naturally lead to the following class of inhomogeneous models, where we start from a homogeneous hamiltonian, and simply vary its hamiltonian density with position. Yet, such deformations can be made for *any* hamiltonian, and we do not need to refer to either SU(2) symmetry or supersymmetry, or any other symmetry, any longer (although, as we just discussed, such symmetries may be a strong motivation to look into this specific class of Hamiltonians). Making the coupling strength position-dependent is a very simple modification that leads to many other examples of inhomogeneous Luttinger liquids with constant parameter $K$. Namely, given a homogeneous Hamiltonian $H$ with hamiltonian density $h_x$,

$$H = \sum_x h_x, \tag{22}$$

which is known to be described at large distance by the (homogeneous) Luttinger liquid model with velocity $v$ and parameter $K$, we can artificially vary the hamiltonian density

$$H \to H[f] = \sum_x f(x)h_x, \tag{23}$$

for some slowly-varying function $f(x) > 0$, and this will give an inhomogeneous Luttinger liquid with position-dependent velocity $v(x) = f(x)v$. We will study this class of models in more detail in section 4.

The rainbow chain [58–60] is a prototypical example of these models with artificially varying Hamiltonian density, which indeed has been already studied with the curved space-time approach [45]. Models with modulated hamiltonian density have also appeared in Refs. [69–73].

## 3 A concrete microscopic example: the Tonks-Girardeau gas in a harmonic trap

For simplicity, we start with a very well-known exactly solvable model: impenetrable bosons in a harmonic trap, defined by the Hamiltonian

$$H = \int dx \left[ \Psi^\dagger \left( -\frac{1}{2}\partial_{x_j}^2 + V(x_j) - N \right) \Psi + c\,\Psi^{\dagger 2}\Psi^2 \right], \qquad V(x) = \frac{x^2}{2}, \tag{24}$$

with

$$c \to +\infty, \tag{25}$$

---

[1]We thank Liza Huijse for pointing that out to us.

and we calculate the boson propagator $\langle \Psi^\dagger(x,t)\Psi(x',t')\rangle$, at different times. We compare the prediction from the effective Luttinger liquid description, valid at large scales, to the exact solution.

## 3.1 Prediction from inhomogeneous Luttinger liquid approach

In the ground state, the average density is given by Wigner's semi-circle law

$$\langle \rho(x)\rangle = \frac{1}{\pi}\sqrt{2N - x^2}, \tag{26}$$

for a large number of particles $N \gg 1$. The position-dependent velocity of low-energy excitations is then

$$v(x) = \pi\langle \rho(x)\rangle = \sqrt{2N - x^2}, \tag{27}$$

and the metric of the effective field theory that describes those low-energy excitations is $ds^2 = dx^2 + v(x)^2 d\tau^2$. As explained in section 2.3, it is more convenient to work with the stretched spatial coordinate

$$\tilde{x} = \int_{-\sqrt{2N}}^{x} \frac{du}{v(u)} = \arccos(-x/\sqrt{2N}), \tag{28}$$

which represents the time needed by a low-energy excitation emitted at the left boundary of the system to arrive at point $x$. The time needed to travel from the left to the right boundary is $\tilde{L} = \int_{-\sqrt{2N}}^{\sqrt{2N}} \frac{du}{v(u)} = \pi$.

The one-particle density matrix in imaginary time was calculated recently by Y. Brun and one of us in Ref. [46]. The expression at different times was not the primary focus of that reference, but it appeared as an intermediate step in the derivation of the main result, see Eq. (35) in [46]:

$$\langle \Psi^\dagger(x,\tau)\Psi(x',\tau')\rangle = \frac{|C|^2}{\sqrt{\pi}}\left|\frac{dw}{dz}\right|^{\frac{1}{4}}\left|\frac{dw'}{dz'}\right|^{\frac{1}{4}}\frac{|w - \overline{w}|^{\frac{1}{4}}|w' - \overline{w}'|^{\frac{1}{4}}}{|w - w'|^{\frac{1}{2}}|w - \overline{w}'|^{\frac{1}{2}}}, \tag{29}$$

where $w = e^{i\pi(\tilde{x}+i\tau)/\tilde{L}}$ and $|C|^2$ is a numerical constant whose exact value was obtained in Refs. [61, 62]: $|C|^2 = G^4(3/2)/\sqrt{2\pi} \simeq 0.521414$. After a few simplifications, and injecting the explicit formula for $\tilde{x}$ for the harmonic trap, this becomes

$$\langle \Psi^\dagger(x,\tau)\Psi(x',\tau')\rangle = \frac{|C|^2}{\sqrt{2\pi}}\frac{\langle \rho(x)\rangle^{1/4}\langle \rho(x')\rangle^{1/4}}{\left[x^2 + x'^2 - 2xx'\cosh(\tau - \tau') + 2N(\cosh^2(\tau - \tau') - 1)\right]^{1/4}}, \tag{30}$$

which is a non-trivial generalization, for $\tau \neq \tau'$, of the equal time correlation in a harmonic trap [63, 64]. Now let us perform the Wick rotation $\tau \to it$. Without loss of generality, we fix $t' = 0$, and write $x_0$ instead of $x'$. This gives

$$\langle \Psi^\dagger(x,t)\Psi(x_0,0)\rangle = \frac{|C|^2}{\sqrt{2\pi}}\frac{\langle \rho(x)\rangle^{1/4}\langle \rho(x_0)\rangle^{1/4}}{\left[x^2 + x_0^2 - 2xx_0\cos t - 2N\sin^2 t\right]^{1/4}}$$

$$= \frac{|C|^2}{\sqrt{2\pi}}\frac{\langle \rho(x)\rangle^{1/4}\langle \rho(x_0)\rangle^{1/4}}{\left[(x - x^+(t))(x - x^-(t))\right]^{1/4}}, \tag{31}$$

where

$$x^\pm(t) = x_0\cos t \pm \sqrt{2N - x_0^2}\sin t, \tag{32}$$

is one of the two positions reached by the signal emitted from $(x_0, 0)$ at time $t$ —one position is reached by the signal propagating to the right, the other by the signal propagating to the left. Straightforwardly, an expression for the different time correlation of hard-core anyons can be obtained generalizing the recent result for equal times [65].

Color density plots as a function of both position and time are shown in Fig. 2. The microscopic and CFT results are shown side by side, and display the same curved lightcones, given by (31). The calculation in the microscopic model is explained in the next section.

## 3.2 Numerical check

The Tonks-Girardeau gas can be mapped onto a system of free fermions. The evaluation of correlations functions in such systems boils down to the computation of determinants [66–68] (or Pfaffians if there are pairing terms, which is not the case here). Such a general observation is a consequence of Wick's theorem [67], which applies for any correlation in a fermionic state, even at finite temperature.

Here we use this to derive exact determinant formulas for the time-dependent one-particle density matrix, which we express as a $(N + 1) \times (N + 1)$ determinant for a system of $N$ particles. The only slight difficulty lies in the fact that the operator that creates a particle at position $x$, $\Psi^\dagger(x)$ is bosonic. The problem can be circumvented by using the second quantized formalism, and attaching a Jordan-Wigner string to a fermionic operator, as is routinely done to diagonalize spin chains such as the XX or Ising chain. More precisely, we write

$$\Psi^\dagger(x) = e^{i\pi\hat{N}(x)}\psi^\dagger(x) = \psi^\dagger(x)e^{i\pi\hat{N}(x)}, \tag{33}$$

where $\psi^\dagger(x)$ is fermionic operator, that satisfies the canonical anticommutation relations $\{\psi(x), \psi^\dagger(x')\} = \delta(x - x')$. The operator $\hat{N}(x)$ counts the number of fermions in the interval $(-\infty; x)$, $\hat{N}(x) = \int_{-\infty}^{x-0^+} \psi^\dagger(z)\psi(z)\,dz$. In such a language the ground state wave function simply reads[2]

$$|\Omega\rangle = \psi_1^\dagger \psi_2^\dagger \ldots \psi_N^\dagger |0\rangle, \tag{34}$$

where $|0\rangle$ is the fermion vacuum annihilated by any $\psi(x)$. The operators

$$\psi_k^\dagger = \int u_k(x)\psi^\dagger(x)\,dx, \tag{35}$$

satisfy the commutation relations $\{\psi_k, \psi_l^\dagger\} = \int_{\mathbb{R}} u_k(x)u_l(x)\,dx = \delta_{kl}$. The $u_k$ are eigenfunctions of the single-particle Schrödinger operator $-\frac{1}{2}\partial_x^2 + V(x)$. When the potential is harmonic the single particle states are known explicitly

$$u_k(x) = \frac{H_{k-1}(x)e^{-x^2/2}}{\pi^{1/4}\sqrt{2^{k-1}(k-1)!}}, \qquad k \geq 1, \tag{36}$$

where $H_p(x)$ is the $p$-th Hermite polynomial. The associated single particle energies are determined from $[H, \psi_k^\dagger] = \epsilon(k)\psi_k^\dagger$, with $\epsilon(k) = k - 1/2$. Let us now write down the time-dependent one-particle density matrix as a fermionic correlator. Using operators in Heisenberg representation $\Psi^\dagger(x, t) = e^{iHt}\Psi^\dagger(x)e^{-iHt}$ and (33) the density matrix reads

$$g_1(x, t, x', t') = \langle\Omega|\Psi^\dagger(x, t)\Psi(x', t')|\Omega\rangle \tag{37}$$

$$= \left\langle 0 \left| \psi_1 \ldots \psi_n e^{iHt}\psi^\dagger(x)e^{i\pi\hat{N}(x)}e^{-iH(t'-t)}e^{-i\pi\hat{N}(x')}\psi(x')e^{-iHt'}\psi_1^\dagger \ldots \psi_n^\dagger \right| 0 \right\rangle.$$

---

[2]In first quantization language the wave function is given by the following Slater determinant $\langle 0|\psi(x_1)\ldots\psi(x_N)\psi_1^\dagger\ldots\psi_N^\dagger|0\rangle = \det_{1\leq j,k\leq N}\left(\langle 0|\psi(x_j)\psi_k^\dagger|0\rangle\right) = \det_{1\leq j,k\leq N}(u_k(x_j))$.

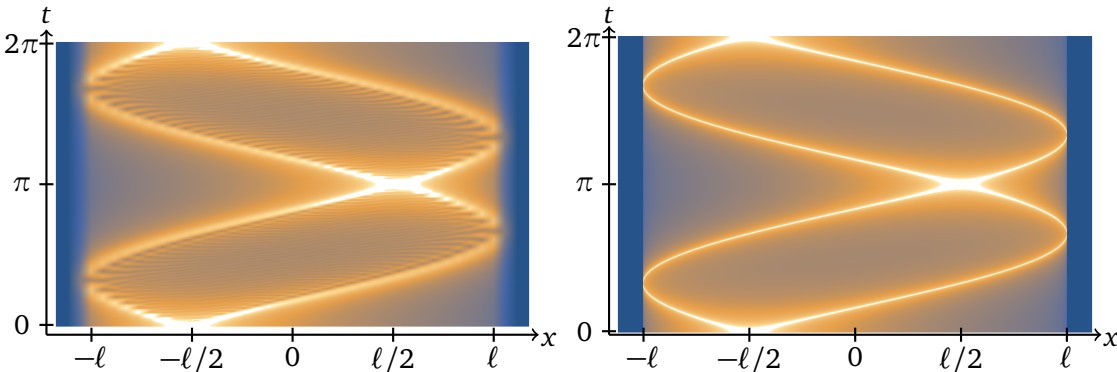

Figure 2: Density plot of $g_1(-\ell/2, t, x, 0)$ as a function of $x$ and $t$. *Left:* Numerics for $N = 160$ particles, with trap size $2\ell = 2\sqrt{N}$. *Right*: CFT formula (31).

Now introduce the ancillary fermion operators

$$\chi(x') = e^{i\pi\hat{N}(x)}e^{-iH(t-t')}\psi(x')e^{iH(t-t')}e^{-i\pi\hat{N}(x)}, \tag{38}$$

$$\chi_k^\dagger = e^{i\pi\hat{N}(x)}e^{-iH(t-t')}e^{-i\pi\hat{N}(x')}\psi_k^\dagger e^{i\pi\hat{N}(x')}e^{iH(t-t')}e^{-i\pi\hat{N}(x)}, \tag{39}$$

to rewrite the above correlator as

$$g_1(x, t, x', t') = e^{-i(t-t')N^2/2}\left\langle 0 \left| \psi_1 \dots \psi_n \psi^\dagger(x)\chi(x')\chi_1^\dagger \dots \chi_n^\dagger \right| 0 \right\rangle. \tag{40}$$

Now we finally make use of Wick's theorem:

$$\left\langle 0 \left| \psi_1 \dots \psi_n \psi^\dagger(x)\chi(x')\chi_1^\dagger \dots \chi_n^\dagger \right| 0 \right\rangle = -\det \begin{pmatrix} 0 & \langle\chi(x')\chi_1^\dagger\rangle & \dots & \langle\chi(x')\chi_n^\dagger\rangle \\ \langle\psi_1\psi^\dagger(x)\rangle & \langle\psi_1\chi_1^\dagger\rangle & \dots & \langle\psi_1\chi_n^\dagger\rangle \\ \langle\psi_2\psi^\dagger(x)\rangle & \langle\psi_2\chi_1^\dagger\rangle & \dots & \langle\psi_2\chi_n^\dagger\rangle \\ \vdots & \vdots & \ddots & \vdots \\ \langle\psi_n\psi^\dagger(x)\rangle & \langle\psi_n\chi_1^\dagger\rangle & \dots & \langle\psi_n\chi_n^\dagger\rangle \end{pmatrix}, \tag{41}$$

where the averages are taken in the fermion vacuum. The matrix elements are given by

$$\left\langle \psi_k\psi^\dagger(x) \right\rangle = u_k^*(x), \tag{42}$$

$$\left\langle \chi(x')\chi_k^\dagger \right\rangle = u_k(x'), \tag{43}$$

$$\left\langle \psi_k\chi_l^\dagger \right\rangle = \int dz\,dz'\,\text{sign}(z-x)\,\text{sign}(z'-x')u_k^*(z)u_l(z')\left\langle \psi(z)e^{-iH(t-t')}\psi^\dagger(z') \right\rangle. \tag{44}$$

The determinant formula (40,41) generalizes the determinant results of Refs. [61, 63] to unequal times $t \neq t'$. We also note that the formula holds for arbitrary potential, provided the $u_k(x)$ solve the corresponding single-particle Schrödinger equation.

While an asymptotic evaluation of this formula in the limit $N \to \infty$ is tedious and would probably require advanced techniques (see e.g. [74] for the uniform case), numerical evaluation for large but finite $N$ is straightforward. In practice the time-consuming step is to evaluate (44), which we rewrite as

$$\left\langle \psi_k\chi_l^\dagger \right\rangle = \sum_{p=1}^\infty e^{-i(p-1/2)(t-t')}\alpha_{kp}(x)\alpha_{lp}(x'), \tag{45}$$

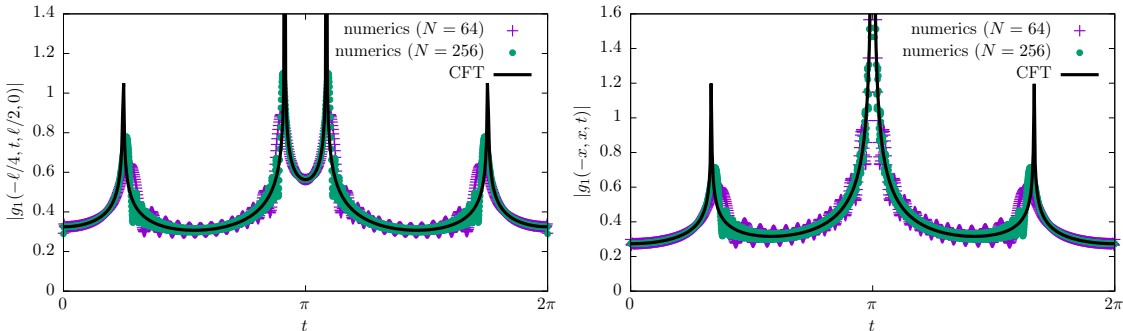

Figure 3: *Left:* One-particle density matrix $|g_1(-\ell/4, t, \ell/2, 0)|$ as a function of time. Here $\ell = \sqrt{2N}$, so that the effective system width is $2\ell$. Numerical computations for $N = 64$ and $N = 256$ are compared to the CFT formula (31). The agreement is very good, and improves further when increasing $N$. *Right:* Same for $|g_1(-\ell/2, t, \ell/2, 0)|$.

with

$$\alpha_{kp}(x) = \int_{-\infty}^{\infty} \text{sign}(z - x) u_k(z) u_p(z) \, dz = \delta_{kp} - 2 \int_{-\infty}^{x} u_k(z) u_p(z) \, dz. \tag{46}$$

The $\alpha_{kp}(x)$ may be evaluated to arbitrary precision using a gaussian quadrature method, and the infinite sum in (45) truncated to some value $p_{\max}$ much larger than the Fermi level $p = N$. Two comparisons between the CFT and exact lattice formula for a finite number of particles are shown in Figs. 2 and 3. In Fig. 3, we fix the two position $x, x'$ to some values and look at the evolution with time. The agreement is very good, and improves further when increasing the number of particles.

## 3.3 Cut-and-glue protocol at the center of the harmonic trap: light-cones and evolution of entanglement entropy

Another way of providing compelling evidence for light-cone propagation of signals is to calculate the evolution of the entanglement entropy after a local quench. Here we focus on the "cut-and-glue" protocol [21], which consists in preparing the system in the ground state of the Hamiltonian

$$H_{\text{cut}} = H + \int dx V_{\text{cut}}(x) \Psi^\dagger(x) \Psi(x), \tag{47}$$

where we imagine that $V_{\text{cut}}(x)$ is a Dirac delta $V_{\text{cut}} = V_0 \delta(x)$ at the center of the trap, with very large amplitude $V_0 \to +\infty$, such that no tunneling is allowed between the left and right half-systems. Then, at $t > 0$, the infinite barrier $V_0 \delta(x)$ is switched off, and we let the system evolve with the original hamiltonian $H$. This "cut-and-glue" protocol is standard in the literature, and is viewed as a small, localized, perturbation of the system at position $x_0 = 0$. It has been largely studied with CFT methods in the homogeneous situation [21, 23]. Following the notations of the previous section, we anticipate that the positions

$$x^\pm(t) = \pm\sqrt{2N} \sin t, \tag{48}$$

will play an important role.

While it is possible to calculate the expectation values of local operators after this local quench [21, 75], here we focus on the time evolution of the entanglement entropy. The calculation of the entanglement entropy follows from Ref. [23]—see Eqs. (34)-(39) in that reference—. We need to evaluate the one-point function of a certain primary operator (called

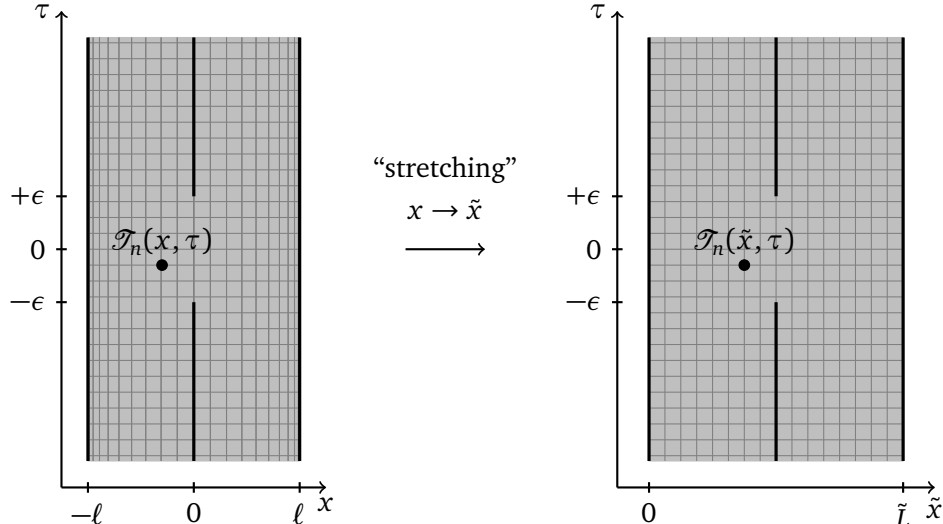

Figure 4: The euclidean spacetime geometry needed for the calculation of the evolution of the entanglement entropy after a "cut-and-glue" quench in the center of a harmonic trap (left). The system is inhomogeneous, with a spacetime metric $ds^2 = dx^2 + v(x)^2 d\tau^2$. In stretched coordinates $\tilde{x}$ (right), the metric is $ds^2 = e^{2\sigma}[d\tilde{x}^2 + d\tau^2]$ with $e^{\sigma(x)} = v(x)$, so it can be made flat with the Weyl transformation $g_{\mu\nu} \to e^{-2\sigma} g_{\mu\nu}$. This is the trick on which we rely to adapt the calculation of the one-point function $\langle \mathcal{T}_n(\tilde{x}, \tau) \rangle$ from Ref. [23] to this inhomogeneous problem.

twist operator [24–27], coming from the implementation of the replica trick in CFT [76]), noted $\mathcal{T}_n(\tilde{x}, \tau)$, in the euclidean space-time geometry shown in Fig. 3.3, in the metric $ds^2 = dx^2 + v(x)^2 d\tau^2 = e^{2\sigma}[d\tilde{x}^2 + d\tau^2]$ with $e^{\sigma(x)} = v(x)$. The first step is to relate this one-point function in curved space to the one in flat space, with a Weyl transformation [43]

$$\langle \mathcal{T}_n(\tilde{x}, \tau) \rangle_{\text{curved}} = e^{-\sigma \Delta_n} \langle \mathcal{T}_n(\tilde{x}, \tau) \rangle_{\text{flat}}, \tag{49}$$

where $\Delta_n = \frac{c}{12}(n - 1/n)$ is the scaling dimension of the twist field $\mathcal{T}_n$ [76]. Then we use a result of Ref. [23]: the formula for $\langle \mathcal{T}_n(\tilde{x}, t) \rangle_{\text{flat}}$, obtained after performing the Wick rotation $\tau \to it$, is given by Eq. (38) in that reference. In our notations, it reads

$$\langle \mathcal{T}_n(\tilde{x}, t) \rangle_{\text{flat}} = \left( \frac{\tilde{L}/\pi}{\cosh \frac{\pi \epsilon}{\tilde{L}} \sinh \frac{\pi \epsilon}{\tilde{L}}} \left| \cos^2\left( \frac{\pi(t + i\epsilon)}{\tilde{L}} \right) - \sin^2\left( \frac{\pi \tilde{x}}{\tilde{L}} \right) \right|^{\frac{1}{2}} \right.$$

$$\left. \times \left| 2 \left| \cos^2\left( \frac{\pi(t + i\epsilon)}{\tilde{L}} \right) - \sin^2\left( \frac{\pi \tilde{x}}{\tilde{L}} \right) \right| - \cos \frac{2\pi \tilde{x}}{\tilde{L}} \cosh \frac{2\pi \epsilon}{\tilde{L}} - \cos \frac{2\pi t}{\tilde{L}} \right|^{\frac{1}{2}} \right)^{-\Delta_n}. \tag{50}$$

Here $\epsilon$, which has the dimension of a time, comes from the definition of the spacetime geometry shown in Fig. 3.3, and it plays the role of a UV cutoff. By dimensional analysis, we expect it to scale as $\epsilon \sim \frac{1}{\langle \rho \rangle v}$ in the center of the trap, which gives $\epsilon \sim 1/N$. When we analyze the behavior of that expression when $\epsilon \to 0$, we find that we need to distinguish two cases:

$$\langle \mathcal{T}_n(\tilde{x}, t) \rangle_{\text{flat}} \underset{\epsilon \to 0}{=} \begin{cases} \left( \frac{2\tilde{L}^2}{\pi^2 \epsilon} \left| \cos \frac{\pi t}{\tilde{L}} - \sin \frac{\pi \tilde{x}}{\tilde{L}} \right| \left| \cos \frac{\pi t}{\tilde{L}} + \sin \frac{\pi \tilde{x}}{\tilde{L}} \right| \right)^{-\Delta_n} & \text{if} \quad \cos^2\left( \frac{\pi t}{\tilde{L}} \right) < \sin^2\left( \frac{\pi \tilde{x}}{\tilde{L}} \right), \\ \left( \frac{\tilde{L}}{\pi} \left| \sin \frac{2\pi \tilde{x}}{\tilde{L}} \right| \right)^{-\Delta_n} & \text{if} \quad \cos^2\left( \frac{\pi t}{\tilde{L}} \right) > \sin^2\left( \frac{\pi \tilde{x}}{\tilde{L}} \right). \end{cases}$$

$$\tag{51}$$

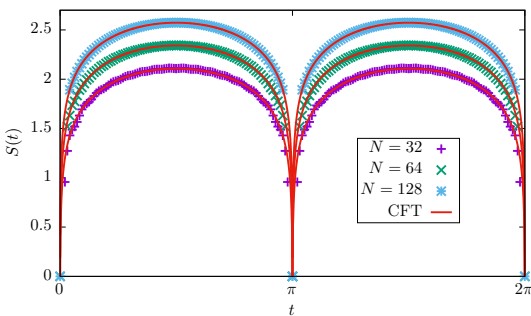
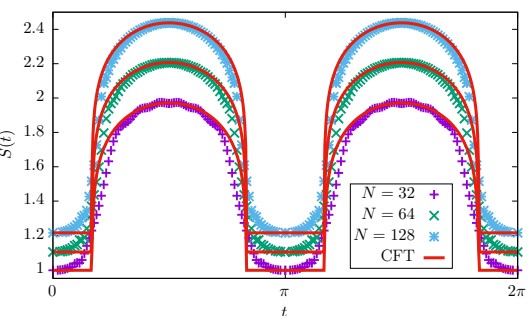

Figure 5: Entanglement entropy as a function of time after a cut and glue quench in the harmonic trap. *Left:* measure in the middle, $x = 0$. As can be seen the agreement between the microsopic calculation and the CFT result is excellent. The agreement even improves for a larger number of particles. *Right:* measure at $x = \ell/2$, for a trap $x \in [-\ell, \ell]$. The agreement is very good, but there are, as discussed in the text, small but noticeable deviations.

Now let us relate this to the entanglement entropy. In CFT, the Renyi entropy is given by $S_n(x, t) = \frac{1}{1-n} \log\left[ \langle \mathscr{T}_n(\tilde{x}, \tau) \rangle_{\text{curved}} a_0^{\Delta_n} \right]$, where $a_0$ is some UV length scale. In the Tonks-Girardeau gas, the only microscopic length scale is the inverse particle density $1/\langle \rho(x) \rangle$, so we must have $a_0 \propto 1/\langle \rho(x) \rangle$ up to some numerical constant. It is important to emphasize that, contrary to the homogeneous case, the UV length scale $a_0$ is actually position-dependent, as noted in Ref. [43]. The Renyi entropy is thus

$$S_n(x, t) = \frac{1}{1-n} \log\left[ v(x)^{-\Delta_n} \langle \mathscr{T}_n(\tilde{x}, \tau) \rangle_{\text{flat}} a_0(x)^{\Delta_n} \right], \tag{52}$$

with $\langle \mathscr{T}_n(\tilde{x}, \tau) \rangle_{\text{flat}}$ given by Eq. (51). Finally, by specifying $\tilde{x} = \arccos(-x/\sqrt{2N})$, $\tilde{L} = \pi$, $a_0(x) \propto \langle \rho(x) \rangle^{-1} \propto (2N - x^2)^{-\frac{1}{2}}$, and $v(x) = (2N - x^2)^{\frac{1}{2}}$, we arrive at

$$S_n(x, t) = \begin{cases} \frac{n+1}{12n} \log\left[ (2N - x^2) \left| x - x^+(t) \right| \left| x - x^-(t) \right| \right] + \text{cst} & \text{if} \quad x \in [x^-(t), x^+(t)], \\[2ex] \frac{n+1}{12n} \log\left[ 2^{3/2} \sqrt{N} x (1 - \frac{x^2}{2N})^{3/2} \right] + \text{cst}' & \text{if} \quad x \notin [x^-(t), x^+(t)]. \end{cases} \tag{53}$$

where $x^{\pm}(t)$ was given in Eq. (48), and where the additive numerical constants are independent of $N$, the number of particles in the trap. This clearly illustrates the light-cone effect: the entropy stays constant while $x$ is outside the interval $[x^-(t), x^+(t)]$, and it starts growing as soon as the signal reaches the position $x$. Again, what is interesting here, compared to the well-known homogeneous case, is that the light-cone is curved: the propagation speed is not constant, but it varies with position. We now turn to numerical checks of formula (53).

**Numerical checks.** We compare here the CFT result to the microscopic model, where the entanglement entropy may be computed using a variation of the correlation matrix techniques [77–79]. We are dealing here with particle conserving Hamiltonians, and states that are obtained by filling some Fermi sea, $|\Omega\rangle = \prod_{k=1}^N \chi_k^\dagger |0\rangle$, where $N$ is the number of particles and $|0\rangle$ is the Fermion vacuum. For such states the von Neumann entropy of some subsystem $A$ is given by the explicit formula

$$S_A = -\text{Tr}\left[ C_A \log C_A + (1 - C_A) \log(1 - C_A) \right], \tag{54}$$

where $C_A = (C_{kl})_{1 \le k, l \le N}$, is the overlap matrix with elements [78, 79]

$$C_{kl} = \left\langle 0 \left| \chi_k^A \chi_l^{A\dagger} \right| 0 \right\rangle. \tag{55}$$

The $\chi_k^{A\dagger}$ are obtained by real-space projection of $\chi_k^{\dagger}$ onto subsystem $A$:

$$\chi_k^{\dagger} = \int_{\mathbb{R}} dx\, v_k(x)\psi^{\dagger}(x) \quad \longrightarrow \quad \chi_k^{A\dagger} = \int_A dx\, v_k(x)\psi^{\dagger}(x). \tag{56}$$

This result may be used to compute the entanglement entropy after the cut and glue quench, in the harmonic potential. Denote as before by $\psi_k^{\dagger} = \int dx\, u_k(x)\psi^{\dagger}(x)$ the fermionic modes that diagonalize the final hamiltonian, $[H, \psi_k^{\dagger}] = (k-1/2)\psi_k^{\dagger}$, and take an initial state of the form $|\psi(0)\rangle = \chi_1^{\dagger} \dots \chi_N^{\dagger}|0\rangle$. One can then write $|\psi(t)\rangle = \chi_1^{\dagger}(t) \dots \chi_N^{\dagger}(t)|0\rangle$, with

$$\chi_k^{\dagger}(t) = \int_{\mathbb{R}} dx\, v_k(x,t)\psi^{\dagger}(x), \tag{57}$$

Thus, the problem of computing the entanglement entropy is reduced to computing the $N \times N$ overlap matrix $C_A(t)$, with matrix elements

$$[C_A(t)]_{kl} = \int_A dx\, v_k^*(x,t)v_l(x,t), \tag{58}$$

and then evaluating (54), which can be done by simple diagonalization. In the case of the Harmonic potential the single particle wave functions before and after the quench may be expressed in terms of Hermite polynomials, which leads after some algebra to

$$v_k(x,t) = \int_{\mathbb{R}} dz\, v_k(z)K(z,x|t), \qquad K(z,x|t) = \sum_{p=1}^{\infty} e^{i(p-1/2)t}u_p(z)u_p(x). \tag{59}$$

The kernel $K$ is known as the Mehler kernel, and may be written as

$$K(z,x|t) = \frac{e^{it/2}}{\sqrt{-2i\pi\sin t}}\exp\left[i\frac{(z+x)^2}{4}\tan\frac{t}{2} - i\frac{(z-x)^2}{4}\cot\frac{t}{2}\right], \tag{60}$$

which means the overlap matrix may be efficiently computed by performing gaussian quadrature on (59) and (60). The results are shown in Fig. 5. It is evident that the agreement between numerics and CFT prediction is excellent, but there are also some deviations which are easily understood in terms of supersonic propagation of some particles as already similarly pointed out in [28]. Indeed in the Tonks-Giradeau gas, there is no bound on the speed of propagation of information and so, even if subdominant, there are quasi-particles traveling with a velocity larger than the Luttinger liquid one $v(x)$. This is evident from the fact that the entanglement entropy slowly and slightly deviates from the initial value before than predicted by CFT. Anyhow, this is clearly a small subdominant effect and the light-cone dynamics is clearly visible in Fig. 5.

## 4 Models with inhomogeneous hamiltonian density

Let us consider a critical spin chain Hamiltonian with open boundaries, of the form

$$H = \sum_{x=-\ell+1}^{\ell-1} \hat{h}_x, \tag{61}$$

where $\hat{h}_x$ is the Hamiltonian density (an operator acting on a finite number of neighboring sites). We denote by $v_f$ the speed of propagation of low-energy excitations. For free fermionic

system this coincides with the Fermi speed; for more complicated models this needs to be determined a priori.

It is always possible to consider the following deformation

$$H[f] = \sum_{x=-\ell+1}^{\ell-1} f(x/\ell)\hat{h}_x, \tag{62}$$

where $f(u)$ is some smooth non-negative function of $u$. In particular, the so-called sine-square deformation $f(u) = \sin^2 \frac{\pi(u+1)}{2}$ has been the subject of several investigations in the past few years [69–73], as the ground state in that case mimicks that of a chain with periodic boundary conditions. Here we look at generic examples of such deformations, in a sense which we explain in the following subsection. For finite but large $\ell$ these automatically give models with spatially-dependent velocity $v(x) = f(x/\ell)v_f$, that also varies slowly at the lattice scale, and where $K$ is the same as in the original homogeneous chain.

The purpose of this section is to investigate the curved light cone physics that emerges from a non-uniform velocity. Our main example will be an inhomogeneous XXZ spin chain, with Hamiltonian density

$$\hat{h}_x = S_x^1 S_{x+1}^1 + S_x^2 S_{x+1}^2 + \Delta S_x^3 S_{x+1}^3, \tag{63}$$

and $|\Delta| \leq 1$ (gapless regime).

## 4.1 Dynamical correlations

Perhaps the simplest illustration of our ideas is provided by the following dynamical spin-spin correlation function

$$D(x_0, x, t) = \langle S_{x_0}^3 S_x^3(t) \rangle, \tag{64}$$

where $S_x^3(t) = e^{iH[f]t} S_x^3 e^{-iH[f]t}$, and the average is taken in the ground state of the inhomogeneous chain. In a homogeneous setting such correlations are well described by CFT, and the lightcones given by the simple equation

$$x - x_0 = \pm v_f t. \tag{65}$$

In the XXZ spin chain the speed $v_f$ is known exactly from the Bethe-ansatz solution [74]

$$v_f = \frac{\pi \sin\gamma}{2\gamma}, \tag{66}$$

with $\cos\gamma = \Delta$. For the inhomogeneous Hamiltonian $H[f]$ the relevant stretched coordinate then becomes

$$\frac{\tilde{x}}{\ell} = \int_{\frac{x_0}{\ell}}^{\frac{x}{\ell}} \frac{du}{f(u)}. \tag{67}$$

and the lightcone is simply given by

$$\tilde{x} = \pm f(x_0/\ell)v_f t, \tag{68}$$

which is the main result of this section. We have assumed here for simplicity that the stretched coordinate (67) is always well defined. This means we need to exclude deformations that vanish too fast – typically as $(x - x_c)$ or faster – for a certain discrete set of points $x_c$. In that case, the integral defining (67) could become divergent if one of the critical points $x_c$ lies in the interval $[x_0, x]$. Physically this simply means that any signal would take an infinite time

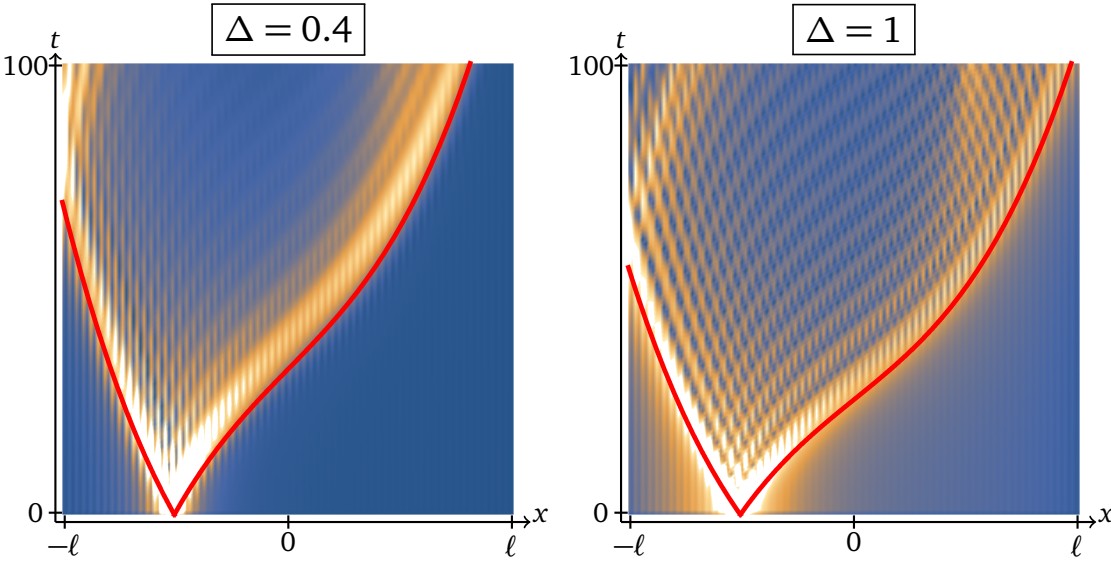

Figure 6: Density plot of the dynamical correlation $\langle S^3_{x_0} S^3_x(t) \rangle$, for $x_0 = -\ell/2$ as a function of position $x$ and time $t$, in XXZ spin chains of length $2\ell = 128$ for $\Delta = 0.4$ (left) and $\Delta = 1$ (right) with $f(u) = \frac{1}{1+3u^2}$. The red thick curves are the CFT predictions (68) for the lightcone. As can be seen they match very well the numerical data.

to go from $x_0$ to $x$. We find it convenient to avoid such situations; in particular, this rules out the sine-square deformation of Refs. [69–72].

A numerical check of formula (68) can be performed as follows. First, find the ground state $|\psi\rangle$ of $H[f]$ using DMRG. Then, apply $S^3_x$ to the ground state, and perform the unitary time evolution. Since applying $S^3_x$ amounts to a local perturbation, the entropy generated after such a "local quench" only grows logarithmically with time, which means relatively large times may be accessed. Then, the overlap of the time evolved state $|\psi(t)\rangle = e^{iHt} S^3_x |\psi\rangle$ with $S^3_{x_0} |\psi\rangle$ may be easily computed. In practice, we used the ITensor C++ library [80] and were able to access times of the order of $t \simeq 100$ for system sizes $2\ell = 128$, which was sufficient to achieve the desired separation of scales, and clearly see the lightcones. The results are shown in Fig. 6 for two different values of $\Delta$ in the gapless regime. As it can be seen the agreement is excellent.

## 4.2 Entanglement entropy after a local quench

Another illustration is provided by the entanglement entropy following the cut and glue quench. Similar to what was done in section. 3.3, we consider the ground state of the Hamiltonian

$$H_{\text{cut}}[f] = \sum_{x=-\ell+1}^{-1} f(x/\ell)\hat{h}_x + \sum_{x=1}^{\ell-1} f(x/\ell)\hat{h}_x, \tag{69}$$

where the left and right parts are disconnected, and then let evolve with Hamiltonian $H[f]$. The CFT formula for this protocol has been derived in Sec 3.3, see in particular Eq. (52). Here we measure the entropy $S(x_0, t, L)$ of a subsystem $[-\ell, x_0]$ as a function of time in this inhomogeneous setting. As explained in section. 3.3, the time-dependence of the entropy in the flat and non flat setting are related by

$$S(x_0, t, \ell) = S_{\text{flat}}(\tilde{x}_0, t, \tilde{\ell}). \tag{70}$$

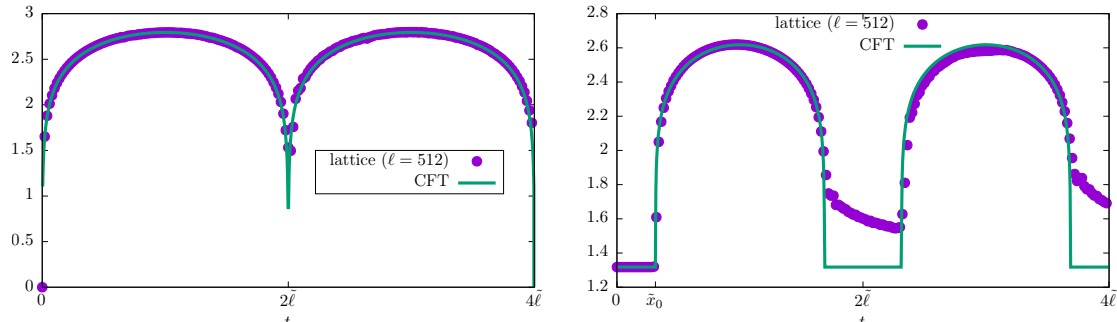

Figure 7: Entanglement entropy after the cut-and-glue quench, and comparison with the CFT formula (71). As in Fig. 6 we chose $f(u) = 1/(1 + 3u^2)$, which implies $\tilde{\ell} = 2\ell$, to make the chain inhomogeneous. *Left:* Cut in the center, $x_0 = 0$, with excellent agreement. *Right:* Cut at $x_0 = \ell/2$, with very good agreement, but slight deviations. Note that the entropy stays constant until time $t = \tilde{x}_0 = 5\ell/8$, yet another illustration of the curved light-cones.

The entropy in the flat (or homogeneous) setting has been derived in [23], which yields

$$S(x_0, t, \ell) = \frac{c}{6} \log \left| \sin^2 \frac{\pi v_f t}{2\tilde{\ell}} - \sin^2 \frac{\pi \tilde{x}_0}{2\tilde{\ell}} \right| + \text{cst.} \tag{71}$$

A numerical check of this formula is shown in Fig. 7 for the inhomogeneous XX chain ($\Delta = 0$), where very large systems and late times may be achieved, and $v_f = 1$. The agreement between the inhomogeneous CFT predictions and the numerics is extremely good and a clear quantitative evidence for the curved light-cone is shown. We must anyhow mention that at a more careful look, very tiny deviations between the two curves can be observed when the measure is not done in the middle ($x_0 \neq 0$, see right panel), similar to what we observed in section 3.3. They do not seem to disappear when increasing the system size, and we found them also in the uniform case. Their origin is unclear, possibly related to issues with the analytic continuation from imaginary to real time, whose subtleties are still not fully understood at present [22, 23]. This effect will be studied elsewhere. Nevertheless, we stress that the analytical prediction for the curved light-cone (no signal until time $t = \tilde{x}_0$), which is the main message of the present paper, is verified to high accuracy, and should become exact in the limit of infinite system size.

## 5 Conclusions

In this paper we have shown that inhomogeneous Luttinger liquids with a uniform Luttinger parameter $K$ at low energy display the universal phenomenon of curved light cones. Gapless excitations propagate along the null geodesics of the metric $ds^2 = dx^2 - v(x)^2 dt^2$. Our approach, based on CFT in curved background [43], allows to quantitatively characterize the curvature of the light-cone and so to determine analytically the explicit space dependence of the sound velocity. We have checked our findings with explicit analytic and numerical calculations both in- and out-of-equilibrium for a Tonks-Girardeau gas in a harmonic potential and in systems with artificially tuned hamiltonian density. We stress however, that there is a large class of inhomogeneous systems which have uniform Luttinger parameter. This includes models with some particular internal symmetry, such as all free fermionic models, model with SU(2) symmetry or supersymmetric ones, as discussed in the text.

The main still open problem is how to generalise our approach to systems with non-uniform

Luttinger parameter, such as 1d Lieb-Liniger model [81] and Fermi gases [82] in a trap potential. For those systems we still lack a complete analytic and quantitative understanding of the propagation of gapless excitations. There seems to be only a very few references that have attacked this problem: the particular case of a step-function $K(x)$ was studied in Refs. [50,51], while partial results where obtained for correlations of the trapped Lieb-Liniger gas in [29]. We believe that this is a fascinating problem on which many physically relevant results remain to be discovered. Work in this direction is in progress and those systems with non-uniform parameter $K$ will be studied elsewhere.

# Acknowledgements

We thank Bruno Bertini, Liza Huijse, and German Sierra for useful discussions, Jacopo Viti for ongoing collaboration on closely related topics, and Yannis Brun for collaboration as well as for his careful reading of the manuscript. The numerical results of section 4 were obtained with the ITensor library [80].

**Funding information**   This work was supported partially by the ERC under Starting Grant 279391 EDEQS (PC), and by the CNRS Interdisciplinary Mission and the Université and Région Lorraine (JD).

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
