# Peer review of "Emergence of curved light-cones in a class of inhomogeneous Luttinger liquids"

_SciPost Physics, doi:SciPost Phys. 3, 019 (2017)_

## Round 2 · Referee Report · Anonymous (Referee 1) · 2017-6-24

Strengths

- Timely subject.
- Analytical predictions supported by numerical evidences.

Weaknesses

- Hard to understand what is really new.
- The implications of the main results are not clear.

Report

The authors study in more details the features of a CFT for Inhomogeneous one-dimensional quantum
systems, where the Luttinger parameter K is nonetheless constant in space. Despite this restriction they show that there are still some models with inhomogenous Hamiltonians (non translationary invariant) that can be described this way. The two examples they provide are free fermions in a harmonic trap (which is exactly solvable) and generic spin chains with space-dependent couplings. Some questions that I have are listed below.

a) Why there is no a general classification of such models with constant K in space? The authors provide two examples but it is not clear how to say a priori if a model fall in their category or not.
b) The authors should stress how different is (and why it is better) their description compared to a standard Luttinger liquid theory with a Local Density Approximation (LDA). Namely one could expect that the same curved light-cone velocities can be predicted by employing a LDA approximation on top of a uniform system.
c) What are the conditions that the function f they introduce in (62) has to obey in order for their approach to be valid? Any smooth (C^\infty) function is ok or there are corrections of order 1/ell^alpha with alpha some number? Could then the authors reproduce a disorder model by considering a function f such that its Fourier coefficients are chosen to be random numbers?
d)The authors show that a curved light cone emerges when observing dynamical correlations. What about a quantum quench or a inhomogenous quench as the ones studied in ref. [35]-[37] ?
e) Is it possible to construct inhomogenous Hamiltonians such that the light cone closes on itself, namely where the particle propagation is confined, as observed in Nature Physics 13 (2017) 246-249?
f) What are the connections of this work with the recent progresses that have been achieved on the hydrodynamic description of integrable model in presence of inhomogenous fields, see SciPost Phys. 2, 014 (2017)?
e) What are the consequences of a curved light cone on transport coefficients like the thermal and charge Drude weight?

Requested changes

I kindly ask the authors to comment about the questions above, mainly in the introduction or in the conclusions.

---

## Round 2 · Referee Report · Anonymous (Referee 2) · 2017-6-26

Strengths

Discussion clear and to the point
Results simple and important
Comparison with numerics very compelling in most cases

Weaknesses

Observed but not fully explained discrepancy between prediction and numerics for entanglement entropy
Discussion of some assumptions
Some references

Report

In this paper, the authors develop further a theory for inhomogeneous critical models. The main point is that, based on very general arguments, models whose large scale physics is described by the Luttinger liquid theory (CFT with central charge $c=1$), when modified to account for external large scale, slowly varying inhomogeneities, should be described by CFT in curved space. This works in a certain class of inhomogeneities, in which the Luttinger liquid parameter is constant. There are various other assumptions that are made in this paper, and explicit models satisfying these assumptions are studied, with very compelling numerical comparison.

I find this paper very nice. It is well written, the discussion is clear, and the results interesting and very well justified, both analytically and numerically. There is numerical discrepancy for the entanglement entropy, which is not fully explained, but I think it is fine to leave this for future work. I just have small points (see "requested changes"), which are nevertheless important before the paper can be published.

Requested changes

1. Assumption 1 is probably quite crucial for the method, at least in its current form, as discussed. This is fine and can be left for future works. However I believe that assumptions 2 and 3 are not that crucial. Could the authors discuss what would happen if assumptions 2 and 3 do not hold? How would the method be modified? Are there natural models where this would occur (e.g. modifying the Hamiltonian density with a space-time dependent factor)?

2. The definition of the stretched coordinate is good, and of course the choice of the initial point $x_0/\ell$ (e.g. in (67)) is somewhat arbitrary. With appropriate choice of initial point, the problem of non-integrability of the integrand with the sine-square deformation could be averted quite easily I believe. Physically, it is just that with an non-integrable deformation, the time to reach the singular point is infinite, hence no excitation can ever reach such a point. This is natural as zeroes of $f$ give zero interaction, hence no propagation through. Thus the region is divided into different regions, between non-integrable points, each with their own stretched coordinate that goes to +/- infinity at the boundaries of the region. Perhaps the authors could discuss this a bit more.

3. This is a reference-related point. In discussing the entanglement entropy, the authors make use of a twist field. I understand this twist field was actually first introduced in the work [J. L. Cardy, O. A. Castro-Alvaredo and B Doyon J. Stat. Phys. 130, 129-168 (2008)] in the context of entanglement entropy, clearly crucially inspired by the first paper in [71]. For instance as far as I'm aware no literature on the entanglement entropy before the above work talks about twist field, while almost all after it refer to this concept. The twist field was also of course studied much earlier in different contexts [e.g. L. Dixon, D. Friedan, E. Martinec and S. Shenker, Nucl. Phys. B675, 13-73 (1987)]. Since the authors actually refer to a twist field and use the notation of the above work, why not cite this literature?

---

## Round 3 · Referee Report · Anonymous · 2017-7-21

Strengths

Discussion clear and to the point
Results simple and important
Comparison with numerics very compelling in most cases

Weaknesses

-

Report

I think the authors have answered well all questions, and changes are appropriate. I think the paper can be published.

Requested changes

none

---

## Round 3 · Referee Report · Anonymous · 2017-7-29

Strengths

-

Weaknesses

-

Report

The authors have addressed all the remarks and they replied to all my questions. I believe the paper is now ready for publication.

---

## Round 3 · List of Changes

Dear Editor,

We wish to thank the referees for their reports, and their positive appreciation of the manuscript.

Below is a point by point answer to the remarks by the referees, together with the changes made to the manuscript.

////////////////////////
Report 176
////////////////////////

"1. Assumption 1 is probably quite crucial for the method, at least in its current form, as discussed. This is fine and can be left for future works. However I believe that assumptions 2 and 3 are not that crucial. Could the authors discuss what would happen if assumptions 2 and 3 do not hold? How would the method be modified? Are there natural models where this would occur (e.g. modifying the Hamiltonian density with a space-time dependent factor)?"

True, assumption 2 and 3 are in principle not crucial for our approach to work. They are however very convenient, as they simplify computations considerably. As the referee points out, assumption 3 may be relaxed by considering space-time dependent densities. Assumption 2 may be relaxed e.g. by performing a global quench instead of a local quench. In that case the quench also changes the space-time geometry, which for small local perturbations is already fixed by the ground state. Also, for a global quench many excitations propagate, and disentangling them would be more complicated.

"2. The definition of the stretched coordinate is good, and of course the choice of the initial point x0/ℓ (e.g. in (67)) is somewhat arbitrary. With appropriate choice of initial point, the problem of non-integrability of the integrand with the sine-square deformation could be averted quite easily I believe. Physically, it is just that with an non-integrable deformation, the time to reach the singular point is infinite, hence no excitation can ever reach such a point. This is natural as zeroes of f give zero interaction, hence no propagation through. Thus the region is divided into different regions, between non-integrable points, each with their own stretched coordinate that goes to +/- infinity at the boundaries of the region. Perhaps the authors could discuss this a bit more."

The referee is fully correct in pointing out that the integrability assumption on 1/g(x) is not so important, and that generalization to non integrable singularities is relatively straightforward. This extra assumption is merely technical, it ensures the signal will reach any point in the chain in a finite time. We felt this was the simplest way of illustrating our main idea, curvature of light-cones, while removing any possible ambiguities in all our formulas. We have amended the manuscript to make that point clearer.

"3. This is a reference-related point. In discussing the entanglement entropy, the authors make use of a twist field. I understand this twist field was actually first introduced in the work [J. L. Cardy, O. A. Castro-Alvaredo and B Doyon J. Stat. Phys. 130, 129-168 (2008)] in the context of entanglement entropy, clearly crucially inspired by the first paper in [71]. For instance as far as I'm aware no literature on the entanglement entropy before the above work talks about twist field, while almost all after it refer to this concept. The twist field was also of course studied much earlier in different contexts [e.g. L. Dixon, D. Friedan, E. Martinec and S. Shenker, Nucl. Phys. B675, 13-73 (1987)]. Since the authors actually refer to a twist field and use the notation of the above work, why not cite this literature?"

That is a fair point. We have added citations to the two papers cited by the referee, as well as to two other references by Knizhnik and by Lunin & Mathur.

////////////////////////
Report 174
////////////////////////

"a) Why there is no a general classification of such models with constant K in space? The authors provide two examples but it is not clear how to say a priori if a model fall in their category or not."

Generically, one expects most models to have non-constant K. We provided two classes of examples for which K is not constant (systems that map on free fermions, and systems with modulated hamiltonian density); we also argued that K may be protected by some additional symmetry like $SU(2)$ or supersymmetry. There might exist other examples where K is constant. But to our knowledge no general classification exists. This is perhaps an interesting question for future research.

"b) The authors should stress how different is (and why it is better) their description compared to a standard Luttinger liquid theory with a Local Density Approximation (LDA). Namely one could expect that the same curved light-cone velocities can be predicted by employing a LDA approximation on top of a uniform system."

LDA is definitely a necessary ingredient in our approach. In the Tonks-Girardeau example treated in the paper, we use the density profile obtained from LDA as an input, and then we write a field theory action with a spatially-dependent velocity $v(x)$. This is exactly what we do. We are not sure what the referee means by ``LDA approximation on top of a uniform system’’, and it is also unclear to us what the ``standard Luttinger liquid theory with a Local Density Approximation’’ is supposed to be. We would welcome suggestions of relevant references, if the referee could provide them.

"c) What are the conditions that the function f they introduce in (62) has to obey in order for their approach to be valid? Any smooth (C^\infty) function is ok or there are corrections of order 1/ell^alpha with alpha some number? Could then the authors reproduce a disorder model by considering a function f such that its Fourier coefficients are chosen to be random numbers?"

The only requirement is that the stretched coordinate (see e.g. Eq.(67)) be well defined. As discussed after (68) we only need to assume that the function f be (say) continuous and does not vanish, or vanishes sufficiently slowly. This is explained after Eq. (68). We have also reworded the text a little bit around this point.
It is also possible to consider functions with Fourier coefficients chosen at random, provided the above still holds.

"d)The authors show that a curved light cone emerges when observing dynamical correlations. What about a quantum quench or a inhomogeneous quench as the ones studied in ref. [35]-[37]?"

We would get similar behavior. In fact, we ourselves studied this problem in an earlier paper, see section 4 of Ref [39].

"e) Is it possible to construct inhomogeneous Hamiltonians such that the light cone closes on itself, namely where the particle propagation is confined, as observed in Nature Physics 13 (2017) 246-249?"

It is always possible to confine particles by making the couplings arbitrarily small somewhere in the system. But this is not the internal confinement as in the nature physics mentionned, it is more localisation in space, even if they share some similarities. See also the answer to point 2 by the other referee, and the corresponding changes made to the text.

"f) What are the connections of this work with the recent progresses that have been achieved on the hydrodynamic description of integrable model in presence of inhomogeneous fields, see SciPost Phys. 2, 014 (2017)?"

Even though the two approaches share a certain degree of similarity, there is no clear-cut connection at this stage. However, we agree it would be desirable to understand better the possible relations between the two.

"g) What are the consequences of a curved light cone on transport coefficients like the thermal and charge Drude weight?"

It is not clear to us how our work relates to studies of Drude weights in quantum systems. We feel it is likely there might be no interesting consequences of our results in this direction.

---

## Editorial Decision

published